# Behaviour in Slower-Growing Broilers and Free-Range Access on Organic Farms in Sweden

**DOI:** 10.3390/ani11102967

**Published:** 2021-10-15

**Authors:** Lina Göransson, Stefan Gunnarsson, Anna Wallenbeck, Jenny Yngvesson

**Affiliations:** Department of Animal Environment and Health, Swedish University of Agricultural Sciences (SLU), S-53223 Skara, Sweden; stefan.gunnarsson@slu.se (S.G.); anna.wallenbeck@slu.se (A.W.); jenny.yngvesson@slu.se (J.Y.)

**Keywords:** welfare, chicken, environmental enrichment, avoidance distance test, slow-growing, predation

## Abstract

**Simple Summary:**

Outdoor access, environmental enrichment and more slower-growing hybrids are means to improve broiler welfare in organic production. Two slower-growing hybrids are currently reared on organic farms in Sweden, but knowledge of bird welfare is limited. Therefore this study surveyed chicken behaviour, including free-range use and features of this, on Swedish organic farms. The results showed that, even towards the end of their production cycle, the chickens were agile enough to ascend various objects for perching. The birds were highly motivated to do so and were provided with a variety of items for perching across farms, but the quantity appeared to be insufficient. On average, almost half of all birds observed on the floor, were in a sitting posture. Free-range areas generally lacked sufficient vegetation cover or artificial shelters, and chickens were mainly observed ranging close to the house. This is novel information on the behaviour and free-range use of two slower-growing hybrids on Swedish organic farms. Key improvements to the indoor environment (e.g., environmental enrichment) and outdoor environment (e.g., vegetation or artificial shelter) could increase broiler welfare. Further research should explore feasible ways for farmers to implement such measures.

**Abstract:**

Two slower-growing hybrids (Rowan Ranger and Hubbard) are currently reared in organic broiler production in Sweden, but knowledge of bird welfare on commercial farms is limited. This study examined chicken behaviour, including free-range use and features of this, in order to enhance knowledge, describe the current situation and identify practical solutions on Swedish organic broiler farms. Eight of 12 available farms were visited once each, when average flock age was 55 ± 6 days. Farmer interviews were followed by avoidance distance tests, group behavioural observations, and assessment of use of environmental enrichment and free-range by the chickens. On average, almost half of all birds observed indoors were in a sitting posture. However, even when approaching slaughter age, the chickens were agile enough to perch and used some of the variety of items provided for perching, but the quantity of environmental enrichment equipment appeared to be insufficient. Free-range areas generally lacked sufficient vegetation cover or artificial shelters, and chickens were predominantly observed ranging near the house. Further research should explore feasible ways for farmers to make key improvements to the indoor and outdoor environment, in order to improve broiler welfare.

## 1. Introduction

One of the fundamentals of organic agriculture is animal welfare [1], which encompasses biological functioning and health, a natural life, and the subjective experience of the animal [2]. In order to promote animal welfare, with emphasis on species-specific behavioural needs, European Union (EU) regulations on organic animal production require e.g., lower stocking densities, at least 8 consecutive hours without artificial light for nocturnal rest and outdoor access for broilers [3]. The latter provides opportunities for important behaviours such as foraging and dust bathing. Although previous research indicates that only a small proportion of a flock uses the free-range areas provided [4,5,6,7], more recent findings based on individual tracking systems demonstrate dynamic ranging behaviour within flocks and indicate that in fact most birds range [8,9,10]. However, studies also show reluctance among birds to venture too far from the house [5,6,11], with some birds rarely or never entering the free-range area [12]. Vegetation cover and artificial shelters can encourage ranging in broilers [5,6,11,13], and one or the other must thus be provided in outdoor areas on organic farms [3,14].

Appropriate breeds should be used within organic production in order to promote animal health and well-being [15]. The rapid growth rate in fast-growing broilers, which is associated with severe welfare issues such as lameness, makes them unsuitable for the longer rearing period in organic production [16,17,18]. Consequently, these hybrids have on organic farms been replaced by more slower-growing broilers with e.g., improved leg health [16,17,18,19,20]. Combined with provision of environmental enrichment (EE) items on organic farms [14], this enables the more slower-growing birds to perform highly motivated and important poultry welfare behaviours [16,19], such as foraging, perching and dust bathing [21].

Swedish regulations, supported by EU organic regulations, allow a maximum average daily weight gain of 45 g for slower-growing hybrids [22]. The two slower-growing hybrids Rowan Ranger^®^ [23] and Hubbard^®^ [24] became commercially available in Sweden in 2014 and 2016, respectively. As a consequence, the number of organic broiler farms increased rapidly between 2015 and 2017, but only 1% of the total Swedish broiler production was organically certified by 2019 [25]. At the time of the study, organic broiler production in Sweden was hence a rather recent development, and knowledge of broiler welfare in the two aforementioned hybrids was limited. Much of the research reported on such hybrids (e.g., [4,6,13,16,17,18,19,20,26]) seems not to have been performed under commercial settings or under conditions compatible with those in Scandinavia. The aim of this study was thus to extend the limited knowledge of organic broiler behaviour, identify details of the free-range area and describe the present situation and practical solutions applied on organic broiler farms in Sweden.

## 2. Materials and Methods

The study did not involve any invasive treatment of the birds observed on commercial farms, merely behavioural observations, and therefore ethical approval by an ethics committee for animal experiments was not required under Swedish legislation [27].

### 2.1. Farms and Flocks

Eight organic broiler farms, all found in the southern third of Sweden, were visited during October (i.e., autumn) 2018. The farmers of all, at the time of the study, established organic broiler farms in Sweden (*n* = 12) were contacted by telephone and informed about the project, and thereafter asked to participate in the study. The farmers successfully contacted and consenting to participate were included and subsequently visited. All participating farms were certified according to KRAV^®^ (Swedish organic incorporated association) standards [14]. Each farm was visited during one day, between 09.30–10.00 and 15.00–15.30, except one farm in which the visit commenced at 08.00 for logistical reasons. All farms were visited by the very same researcher (L.G.) and an assistant. One flock per farm was observed. The farm visits were performed as close to the time of slaughter as possible, and the broilers were on average 55 ± 6 (mean ± SD) days of age at the time. Rowan Ranger and Hubbard JA57/Hubbard JA87, both slower-growing hybrids, were reared in mixed-sex flocks on five and three farms, respectively. The former received eggs (Rowan Ranger) for on-farm hatching, while the latter received day-old (Hubbard) chicks. The chicks were kept in specific arrival compartments until around three weeks of age, when they were moved to a rearing compartment. Average flock size was 4217 ± 1290 and average farm size was 8975 ± 1688 (mean ± SD). Pop-holes connected the rearing compartment to a winter garden (a roofed platform with three walls and one wind net, littered floor and natural ventilation) from where the birds accessed the free-range area. On one farm, chickens were temporarily reared in eight formerly used mobile houses, due to ongoing reconstruction work in the otherwise used rearing compartments. On all farms, the birds were provided with roughage, objects to perch on, natural light inlets and at least eight consecutive hours of nocturnal rest, in accordance with current EU regulations [3] and KRAV^®^ standards [14]. A more detailed description of the farms, flocks and housing can be found in a previous publication by our research group [28].

#### 2.1.1. Farmer Interviews

During the farm visits, all farmers (i.e., bird caretakers) were interviewed according to a structured protocol (Appendix A) about management and husbandry routines, housing, bird health and behaviour (including free-ranging), productivity and free-range features. 

#### 2.1.2. Indoor Observations

Following the interview with the farmer, observations were performed in one broiler flock in one rearing compartment. The flock to observe, in case of two flocks of similar age, was selected by the farmer. The findings on housing system, indoor environment and bird health can be found in our previous publication [28].

An avoidance distance test (ADT) was performed according to the Welfare Quality^®^ assessment protocol for poultry (WQ) [29] at each of the following five locations in the rearing compartment: entry door, adjacent water line in the centre of rearing compartment, centremost pop-hole, halfway along outer short side, and halfway along inner long side. The protocol used in the study also included number of birds touched (not included in the WQ). Observations were always made in this same order of locations when walking through the rearing compartment, and always at least five minutes after the observer had entered the compartment. The ADT was not continued if no chickens had been touched or counted at arm’s length in these five trials, but was otherwise repeated 21 times in total. On the same five locations, the number of birds using the EE closest to the observer was recorded, along with a description of the EE object. The number of birds positioned on top of and adjacent (sitting or standing on the floor, including birds pecking at object, when applicable) to the EE item were counted.

Scan sampling of the behaviour (Table 1) of birds in and adjacent to (in direct contact with) the pop-holes was performed during five consecutive minutes. The number of birds observed performing different behaviours was recorded at three different pop-holes for each flock. During observations, the observer stood approximately 6 m away from the pop-holes to reduce the risk of affecting bird behaviour, while still having a clear view.

The behaviour of birds in groups inside the rearing compartment was continuously observed during five consecutive minutes. Observations were performed halfway along the inner long side of the house, while the observer was sitting down (Figure 1). For habituation, the observer sat for 5 min prior to these behavioural observations. The time from when the observer sat down until the first bird was within arm’s length and the time until the first bird was touched, were recorded. The total number of birds within arm’s length and the total number touched after 3 and 5 min were counted. For the subsequent behavioural observations, birds within an imaginary semi-circle with radius 5 m were included (Figure 1). State behaviours (i.e., behaviour patterns of relatively long duration [33]) were recorded as estimated proportion (%) of birds performing the behaviour (overall assessment for 5 min), while the number of birds observed performing event behaviours (i.e., behaviour patterns of relatively short duration [33]) was counted.

#### 2.1.3. Outdoor Observations

Outdoor observations were performed during the last part of the farm visits. From a position with a good overview, yet without disturbing the birds, the total number of chickens and the proportion of these ranging at certain distances from the winter garden were estimated (Table 2). The observation area did not include the winter garden.

From a position with a good overview, a panoramic photograph was taken of the entire free-range area. Outdoor air temperature (°C) and humidity (%) were recorded at ground level. Precipitation (mm), wind speed and direction (m/s), and time of sunrise and sunset were recorded using a meteorological software mobile telephone application (Swedish Meteorological and Hydrological Institute, SMHI) [34].

### 2.2. Statistical Analyses

No indices were calculated according to The Welfare Quality^®^ assessment protocol, which was used for scoring only. Microsoft Excel (2016) was used for data compilation and diagram creation. All statistical analyses were performed in R. Results are presented as mean and standard deviation for normally distributed variables, and as median and range for non-normally distributed count variables. 

## 3. Results

### 3.1. Indoor Observations 

#### 3.1.1. Avoidance Distance Test (Fearfulness) 

On the first seven farms visited, no chickens were touched or counted at arm’s length in the five ADT trials at different locations in the rearing compartment, so the test was abandoned on these farms. On the eighth and final farm visited, two birds were touched and one bird was counted at arm’s length in five ADT trials. During the five minutes of habituation prior to behavioural observations, no birds were touched on any of the farms. The median number of birds counted at arm’s length was 1 (range 0–3) and 1.5 (range 0–7) after three and five minutes, respectively. The minimum time for a chicken to approach was on average 108 ± 98 (mean ± SD) seconds (one farm where no birds approached within five minutes excluded).

#### 3.1.2. Behavioural Observations in Rearing Compartment

The average proportions of birds observed performing different state behaviours during behavioural observations in the rearing compartment are presented in Figure 2. The behaviour most commonly observed was sitting, followed by standing and walking. The numbers of birds observed performing different event behaviours are presented in Table 3.

#### 3.1.3. Environmental Enrichment

Environmental enrichment was provided on all farms, as reported by all farmers. On all farms but one, an assortment of different items intended as EE objects was observed in the rearing compartments during farm visits (Figure 3).

Home-made wooden perches were provided on two farms (Figure 4). Total perch length was 21.6 m and 19.5 m, respectively, or 0.51 and 0.46 cm, respectively, per bird present in the house at the time of the visit.

Numbers of observations of birds sitting or standing on and adjacent to the different items are presented in Table 4.

#### 3.1.4. Behavioural Observations at Pop-Holes

The average width of the pop-holes was 170 ± 34 cm and the average height was 46 ± 8 cm (mean ± SD). The numbers of birds observed performing different behaviours at the pop-holes during behavioural observations are presented in Table 5. The behaviours most commonly observed were standing, sitting and foraging.

### 3.2. Outdoor Observations

The chickens were allowed outdoor access at around 23–30 days of age. Most commonly, the chickens had access to the winter garden only during the first 1–2 days. Thereafter, they could typically access the outdoor area from 07.30–08.30 h until darkness. Continuous (i.e., also during the night) access to the free-range or winter garden was however provided in summertime on one and three of the farms, respectively. Free-range access was weather-dependent throughout the year on all farms, and typically provided from spring (March–May) until autumn (September–November). The pop-holes were normally open and the winter garden accessible to the chickens when outdoor temperature was above 0 °C. All flocks had access to the free-range during farm visits.

#### 3.2.1. Free-Range Features

The free-range areas on most farms consisted mainly of pasture, with little or no vegetation cover (Figure 5). Vegetation cover was typically restricted to a particular area of the range. The free-range areas comprised natural (not planted) vegetation on all farms. In general, farmer interviews revealed reluctance to plant trees or bushes, since it would hamper pasture topping or crop production. However, two farmers had considered planting currant and blueberry bushes, respectively. Artificial shelters were provided on five farms (Figure 5), typically within 25 m from the winter garden.

#### 3.2.2. Free-Ranging Behaviour in Chickens

Outdoor observations were performed at 14.00–15.30 on seven farms and at 08.00 on one farm. Time of sunrise and sunset was 06.58 h and 19.03 h, respectively, when visiting the first farm, and 07.56 and 17.42, respectively, when visiting the last farm. The estimated number of chickens observed ranging, along with current weather conditions, are presented in Table 6. On all farms but one, the maximum distance from the winter garden to where a bird was observed corresponded to the point at which there was no more artificial shelters or vegetation cover. Farmers estimated the maximum distance birds ranged from the winter garden to be 50–65 m (seven farms) and 150 m (one farm).

#### 3.2.3. Predation

Three farmers considered ground and/or aerial predators to be a significant problem. Foxes (*Vulpes vulpes*), badgers (*Meles meles*) and birds of prey were mentioned specifically. Two farmers reported occasional problems with aerial predators in particular only. The remaining three farmers reported none or minor problems with ground and/or aerial predators. In terms of fencing as a measure to exclude ground predators, there were large differences between the farms (Table 7).

## 4. Discussion

### 4.1. Indoor Observations

#### 4.1.1. Avoidance Distance Test (Fearfulness) 

No chickens were touched or counted at arm’s length in five ADT trials on seven farms, which may reflect a general fearfulness of humans or specific fearfulness of an unfamiliar human wearing unfamiliar clothing [35,36]. The test was likely not biased by poor leg health [37] or high stocking density [38], biases previously described in studies on fast-growing broilers in conventional production and in experimental conditions, respectively. Most birds (94%) assessed were observed to have no or minor gait impairments (see [28]), and were thus able to distance themselves from the observer. Bird density during farm visits was 17.1 ± 1.9 kg (mean ± SD) per m^2^ (see [28]) and allowed birds to move away without obstruction.

On the eighth farm visited, two birds were touched and one bird was counted at arm’s length in five ADT trials. No more trials were performed, since no more than five trials per flock had been completed on the preceding farms. On the eighth farm, the birds were kept in relatively small mobile houses. This possibly explains why they were less fearful, since in a smaller compartment the distance between farmer and birds at any one time decreases, and visual contact presumably increases, which has been shown to improve the human–bird relationship and reduce chicken fearfulness [39]. 

The average minimum time for a chicken to approach the observer during the habituation period prior to behavioural observations was almost two minutes. During the ADT trials, the observer squatted for only 10 s, according to the WQ protocol, which appears to have been insufficient time for the chickens to begin to approach. 

#### 4.1.2. Behaviour in Rearing Compartment

Sitting was the most common behaviour observed, performed by on average almost 50% of the birds included in the group observations. Most farms were visited near the time of slaughter, and thus body weight and stocking density [17] might explain this inactivity, as slower-growing hybrids also become less active with age [16,19,20]. Although relatively slower-growing hybrids spend less time sitting and inactive in comparison with fast-growing strains [16,17,20], it is important to emphasise that the broilers in this study had average daily weight gain of 45–50 g (see [28]). Thus, as previously noted [18], this growth rate might not have been slow enough to alleviate the effect of weight on chicken behaviour. Furthermore, although no severe lameness was observed, only 23% of the chickens walked without any gait impairment (see [28]). The large proportion of birds observed sitting in this study may partly be explained by minor gait impairments, a correlation recently demonstrated in fast-growing broilers [40]. 

Foraging, which was performed by less than 10% of the birds observed, has also been shown to decrease with age in both fast- and slower-growing hybrids [16,17,19]. The adult red junglefowl (*Gallus gallus*) spends around 60% of its active time foraging [21], while the corresponding time allocated in 9-week-old slower-growing broilers is reported to be less than 5% [16,17]. Foraging is an important species-specific behaviour and, although broilers in commercial production are not required to forage to the same extent as their ancestor in the wild in order to meet their nutritional needs, the relatively small time broilers spend foraging in comparison should be noted. It might be explained by increasing body weight hampering active behaviours [17], but also by poor litter quality, which was observed on some of the farms (see [28]). Although free-ranging provides increased foraging opportunities, it is important that good litter quality is maintained indoors, especially during wintertime when the birds do not have outdoor access. 

Preening, an important comfort behaviour in poultry, was performed by on average 8.5% of the birds observed. Adult red junglefowl spend around 12% of their active time preening [21], but slower-growing broilers have been reported to allocate less than a third of this to preening at 9 weeks of age [17]. The sampling method used in this study did not allow for evaluation of time budgets, but the results indicate that the environment on the farms allowed for comfort behaviours (on the floor) to be performed in a synchronised manner, which is important to poultry [41]. Other comfort behaviours, such as dust bathing, leg and wing stretching and play behaviours, were rarely or never observed, however. The sampling method used and the limited time of recording were likely insufficient [33] to detect any e.g., stretching [42] or play behaviour [32]. Moreover, observations were performed around noon on all farms but one (where observations were performed around 09.00), and it is possible that the time of day reduced the chances of observing these behaviours [17,40,42]. However, the results are in agreement with previous findings for slower-growing hybrids [16,19], and other possible explanations for e.g., the lack of observed dust bathing include poor litter quality [19]. Wood shavings were predominantly used as litter (see [28]), and might have been an undesirable type of dust bathing substrate [43,44]. Further studies are necessary to gain a more profound understanding of the behavioural repertoire and time budgets in these two slower-growing broiler hybrids in a commercial environment.

#### 4.1.3. Environmental Enrichment

The chickens in this study were provided with a variety of different types of EE items. The on-farm observations indicated that providing broilers with EE does not have to be particularly complicated or expensive. However, it is important that the items provided are suitable for the purpose and meet the behavioural needs of chickens [45]. Some of the EE observed, e.g., perches and a cart, were frequently used by the birds. Others, such as a stepladder, were not. Straw bales provided a structure for the broilers to sit on top of and also foraging opportunities, in agreement with previous findings [19]. Chickens were commonly observed tightly clustered around the EE items, which suggests, in agreement with previous studies [19,46], that the chickens used these items as shelter while resting. However, chickens gathering around the EE might also indicate an insufficient space allowance per bird on top of these items. For instance, perch length was approximately 5 m/1000 birds, or 0.5 cm per bird, compared with 18 cm and 20 cm perch per bird required for laying hens and guinea fowl, respectively [3]. This can negatively affect bird welfare, since it is important that perching can be performed synchronised [41], especially during night-time. Chickens were also observed perching on items not primarily intended for perching but, for example, for dust bathing (e.g., plastic troughs or pallet collars), and feed and water lines. This confirms that these slower-growing broilers are both motivated and physically capable of perching [17,20]. Several studies have evaluated the suitability of different items as EE for broilers (for review, see [47]), but research investigating the optimal amounts or distribution is still needed. Furthermore, no minimum requirements on EE quantity are specified in the standards which require organic broilers in Sweden to have access to such structures [14]. The provision of EE must be predominantly based on the behavioural needs of chickens, but for such measures to be implemented these must also be practically and economically feasible for farmers.

#### 4.1.4. Behaviour at Pop-Holes 

The most common behaviour observed at the pop-holes was standing, followed by sitting. Chickens thus seemed to appreciate this position between the outdoor and indoor environments. Previous studies on laying hens have also found that some birds commonly remain sitting in the pop-holes [48,49]. On four of the farms, the pop-holes were quite crowded with birds. This appeared to correlate to (although no statistical analysis was possible) a difference in elevation (range 10–19 cm) between the floor of the rearing compartment and the pop-hole, creating a raised threshold for chickens to ascend. On farms where there was no such raised threshold, the pop-holes were not as crowded. Birds perching in the pop-holes could cause crowding [49], and observations of agonistic behaviours might thus be expected. However, aggressive pecking and fighting were rarely and never observed, respectively, at the pop-holes, in agreement with previous findings for laying hens [50]. These observations may indicate that pop-holes have the potential for inherent value, not only as an entry or exit but for the birds both to make a functional choice and express motivated behaviours. 

### 4.2. Outdoor Observations

#### 4.2.1. Chicken Free-Ranging and Free-Range Features

The proportion of chickens observed ranging at the time of the visit was low in all flocks. On seven of the farms, less than 6% of the flock was observed in the outdoor area. This is in agreement with findings in previous (though predominantly experimental) studies on both fast- [5,7] and slower-growing broilers [4,6] showing that only a small proportion of the flock ranges at any one time. However, individual tracking systems have shown bird ranging to be highly dynamic within a flock throughout the day [8,10]. Thus, counting the number of birds on the free-range area at a particular point in time provides limited information about range use within a flock and throughout the day. Weather conditions might have affected chicken ranging [5,13,48] in this study. In two flocks no chickens were observed outdoors, likely due to heavy rain during one farm visit and windy weather (in addition to no vegetation cover and scanty artificial shelter) during the other [48,51].

On the eighth farm, approximately one-fifth of the flock was seen ranging. Observations on this flock were performed in the morning (as opposed to early afternoon on the other farms) due to logistics, and the flock was considerably smaller in size than the other flocks observed. Both are factors which could account for the larger proportion of birds ranging on this farm [4,5,6,7,10].

The broilers in this study might have been discouraged from ranging due to generally limited protective cover from vegetation and artificial shelters. On four farms, the observed free-range areas contained no vegetation cover, or trees and bushes only sparsely scattered throughout the range. Trees and bushes covered one-fifth or more of the entire free-range area on only two farms. Previous research has shown that natural vegetation cover attracts broilers onto the free-range area [5,11,13], and in fact the three flocks with a higher proportion of chickens ranging had a free-range area with relatively more vegetation cover and the majority of ranging chickens were observed in these areas. Although farmers in general were reluctant to plant trees or bushes, two farmers did consider planting (currant and blueberry bushes, respectively). Such integrated production systems could benefit chickens and farmers, providing the former with protection and the latter with additional income.

Artificial shelters were provided in five of the free-range areas studied, of which four contained no or minimal vegetation cover. However, the total overhead artificial shelter area was limited (at most 28 m^2^). Artificial shelters may encourage ranging in broilers [6], but the number of chickens ranging was too low to evaluate any such effects of the items provided in this study. The effects of similar structures on bird ranging have been studied previously [6,13,51,52], but predominantly in laying hens, and without evident or consistent results. To reach the artificial shelters, the chickens commonly had to cross a barren area adjacent to the winter garden. It has been suggested that this distance, where birds are exposed to, e.g., aerial predators, might undermine the effect of artificial shelters [53]. 

The maximum distance from the house to where at least one chicken was observed ranging was 55 m and, on all farms but one, this distance corresponded to the point where there was no more vegetation cover or artificial shelter. This, and similar findings in previous studies demonstrating reluctance of birds to venture too far from the house [5,6,11], emphasise the importance of providing protective structures in free-range areas. The artificial shelters on farms in the present study were commonly placed no farther than 25 m from the winter garden, which was typically as far as any broiler was observed in free-range areas providing artificial shelter only. Thus, a more even distribution of artificial shelters might encourage these broilers to explore larger areas of the free-range [52,54]. Further studies are needed to investigate this in terms of how to successfully encourage broiler ranging. 

#### 4.2.2. Predation 

Three farmers considered ground predators to be a significant problem. On two of these farms, fences were slack and clearly did not protect against e.g., foxes. Only two farmers reported no problems with ground predators, and theirs were the only farms on which wildlife fences were complemented by an electric fence. However, on one farm the free-range area was not completely enclosed but the farmer did not consider ground predators to be a major problem. It should be emphasised that predator issues were defined in terms of the farmer’s perception, rather than based on information on flock mortality. Thus it must be taken into account that there are likely differences between individual farmers regarding what are considered no, small or severe problems. Nevertheless, since the primary function of the fences is not to keep animals in, but to keep predators out, free-range areas must be equipped with fences suitable for the purpose in order to mitigate what currently poses an important animal welfare issue in free-range poultry systems [55].

### 4.3. Limitations of the Study 

At the time of the study, the eight farms included represented two-thirds of the total number of commercial organic broiler farms in Sweden, to the best of our knowledge. Of the remaining four farms (not included in the study), one declined when asked to participate and three were unsuccessfully contacted. Due to the limited number of farms, it was not possible to separate e.g., the effect of hybrid from other specific farm-related factors. This also created difficulties in analysing e.g., the effect of EE on bird behaviour, due to the uniqueness of EE items used on each farm. The flock to be observed, in the case of two flocks of the same age, was chosen by the farmer. This potential bias could have been avoided by using a simple method for randomisation. The flocks were to be observed as close to the time for slaughter as possible, yet the autumn weather still allowed them to have outdoor access, which led to an age variation between some of the flocks. Ideally farm visits would have been performed around similar ages in all flocks. Farm visits were limited to one day on each farm. Behaviours that are affected by e.g., time of day and current weather conditions are thus difficult to extrapolate further. Repeated observations within a flock and in more than one flock per farm, while not possible in this study due to time limitations and for logistical reasons, would have enabled more profound conclusions to be drawn. However, despite the lack of repeated observations, the results from the bird observations and from farmer interviews provide novel insights into organic broiler production on Swedish farms.

## 5. Conclusions

There is a limited number of on-farm studies of slower-growing broilers in organic production available. This study extended existing knowledge and provided a first, although limited, overview of bird behaviour in two slower-growing hybrids in commercial settings and of the free-range areas on organic broiler farms in Sweden. Behavioural observations showed that even at the very end of their production cycle, the more slower-growing hybrids were agile enough to distance themselves from unknown humans and to ascend various objects for perching. However, a large proportion of the birds were observed in a sitting posture on the floor, indicating that attention should be paid to the effect of current growth rate on the behaviour of these slower-growing broilers. Birds were highly motivated to perch and were provided with a wide variety of items to sit upon, but the quantities of items and space for perching appeared insufficient. In general, the free-range areas lacked sufficient vegetation cover or artificial shelter, and chickens were mainly observed ranging close to the house. Future research should thus aim at identifying key improvements to the indoor and outdoor environment that meet the behavioural needs of the chickens and are also feasible for farmers to implement, in order to increase animal welfare, which is one of the fundamentals of organic animal farming.

## Figures and Tables

**Figure 1 animals-11-02967-f001:**
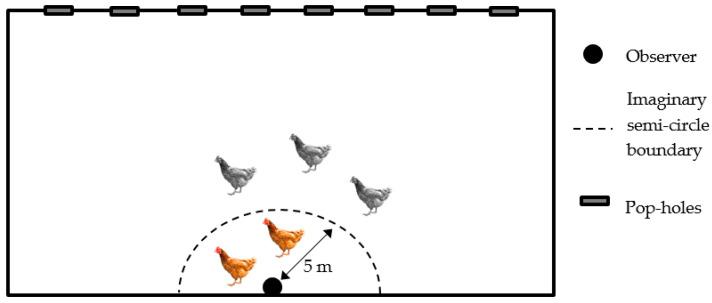
Schematic drawing of imaginary semi-circle boundary used for group behavioural observations of broiler chickens on organic farms.

**Figure 2 animals-11-02967-f002:**
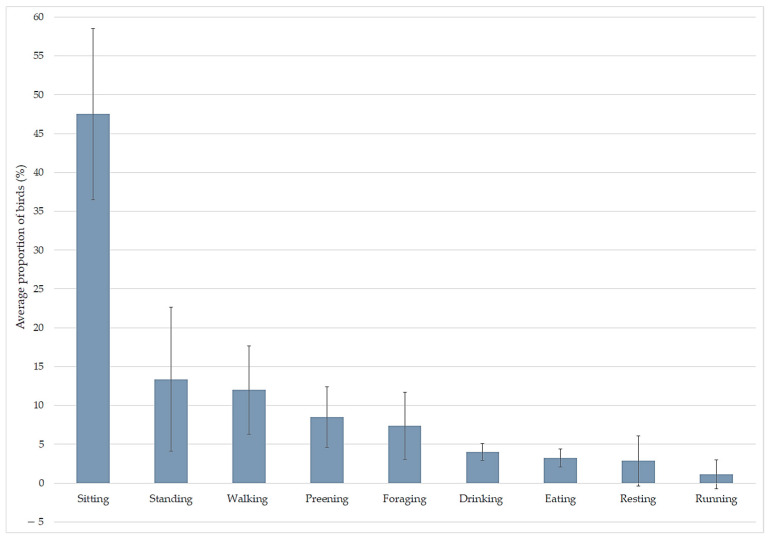
Average proportion of birds observed performing different state behaviours (mean ± SD) during behavioural observations (5 min per farm) on organic broiler farms (*n* = 8 farms).

**Figure 3 animals-11-02967-f003:**
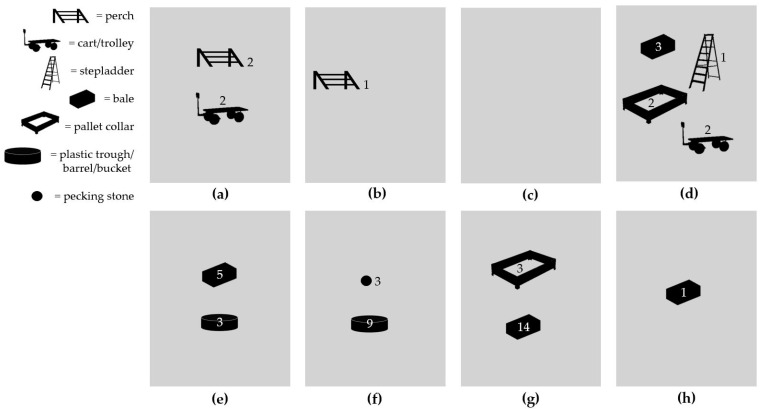
Schematic illustrations (note: not to scale) of environmental enrichment items observed on organic broiler farms (*n* = 8) in Sweden: (**a**) perches and trolleys; (**b**) perch; (**c**) no environmental enrichment; (**d**) Lucerne bales, pallet collars, stepladder and carts; (**e**) peat bales and plastic troughs with dust bathing substrate; (**f**) one plastic trough with dust bathing substrate and eight plastic troughs upside down; (**g**) straw bales and pallet collars with dust bathing substrate; (**h**) one large round straw bale.

**Figure 4 animals-11-02967-f004:**
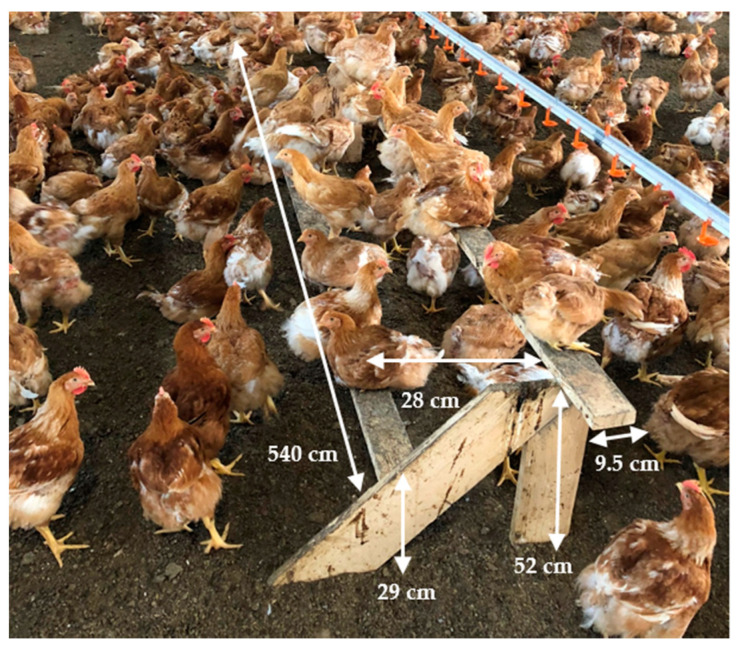
Two-tier home-made wooden perch observed on one organic broiler farm.

**Figure 5 animals-11-02967-f005:**
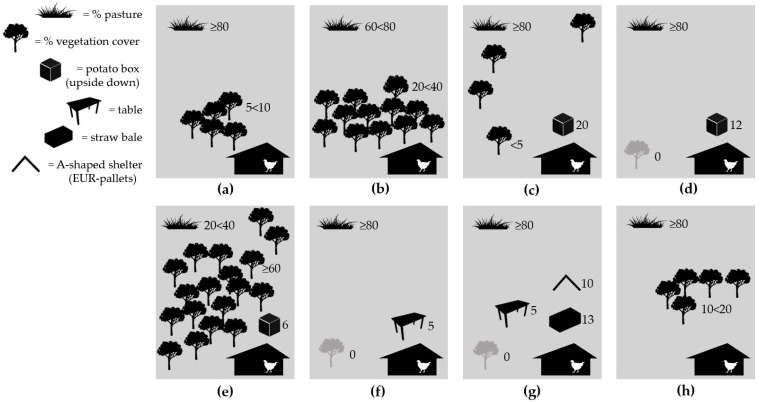
Schematic illustrations (note: not to scale) of vegetation dispersion and artificial shelters (AS) in free-range areas on organic broiler farms (*n* = 8) in Sweden: (**a**) one single distinct area with trees and bushes within 55 m from winter garden; (**b**) population of trees and bushes without clear boundary within 65 m from winter garden; (**c**) six distinct areas (single trees and occasional bushes) distributed throughout the free-range. Total AS overhead cover area 24.6 m^2^; (**d**) pasture only. Total AS overhead cover area 14.4 m^2^; (**e**) trees and bushes distributed throughout the range (difficult to get a clear view of the entire range). Total AS overhead cover area 7.5 m^2^; (**f**) pasture only. Total AS overhead cover area 15 m^2^; (**g**) pasture only. Total AS overhead cover area 28.2 m^2^; (**h**) one single distinct area with trees and bushes within approximately 50 m from house.

**Table 1 animals-11-02967-t001:** Ethogram used for behavioural observations on Swedish organic broiler farms (modified from Rodriguez-Aurrekoetxea et al. (2015) [26] and Ventura et al. (2012) [30]).

Behaviour	Description
States	
Standing	Upright motionless position on extended legs with both feet, but no other body parts touching the ground during ≥2 s
Sitting	Positioned with bent legs, hocks resting on the ground and abdomen in contact with the ground
Resting	Positioned with sternum in contact with the ground, head lowered and resting on ground or tucked in under own wing, with eyes open, semi- or fully closed
Walking	Locomotion starting when bird takes two or more steps forward in succession
Perching	Bird standing, sitting or resting positioned on perch or other elevated structure
Foraging	Bird lowers its head and manipulates substrate on ground with beak and scratches with feet in search of food, while standing or slowly walking forward with head below rump level
Eating	Bird with head above or in feeder, actively consuming feed
Drinking	Bird pecking at drinking nipple or consuming water from cup beneath drinking nipple
Events	
Preening	Manipulation (cleaning, arranging or oiling) of own feathers with beak, while standing or sitting
Dust bathing	Bird sitting or lying down in substrate, pecking and scratching at litter material, tossing and distributing loose substrate onto its back and wings, ruffling and shaking its feathers with or without rubbing head against ground
Wing stretching	Slowly extending one wing
Leg stretching	Slowly extending one leg backwards or laterally
Running	Rapid locomotion starting when bird takes two or more steps forward in rapid succession
Flying	Locomotion starting when bird extends and flaps wings and moves a distance through the air
Gentle feather pecking ^1^	Bird uses beak to gently manipulate and lightly peck at feathers of recipient bird, which does not move away
Severe feather pecking ^1^	Bird uses beak to forcefully manipulate feathers of recipient bird, which moves away from performer bird. Pecks are hard, fast and often singular and may result in detached feathers
Aggressive pecking ^1^	Bird raises head and uses beak to forcefully stab at recipient bird, which moves away. Pecks usually directed towards the head, but may also be directed at the body
Fighting	Two birds standing facing each other, heads and necks raised to the same level, at least one bird forcefully kicking and pecking at conspecific
Pop-hole: walking along ^2^	Bird walking in pop-hole parallel to its opening, at least three steps in succession
Pop-hole: turning back in ^2^	Bird walking or running through pop-hole from inside towards outside, but making a halt and change of direction to remain indoors
Pop-hole: turning back out ^2^	Bird walking or running through pop-hole from outside towards inside, but making a halt and change of direction to remain outdoors
Play-like activity ^3^	Simulated fighting with jumping, kicking and pecking but without obvious aggression or forceful or injurious contact
Sparring ^4^	
Frolicking ^4^	Spontaneous burst of running and/or jumping with wings flapping, with no obvious intention, often with rapid direction changes
Food-running ^4^	Bird picks up an object and runs with it in beak, often making peeping noises repeatedly, followed by at least one other bird
Vocalisations	Sudden loud, sharp, shrill, piercing cry
Squawks	
Other	All other abnormal, aberrant vocalisations

^1^ From Daigle (2017) [31]. ^2^ Pop-hole observations only. ^3^ From Baxter et al. (2019) [32]. ^4^ Recorded as “play-like activity”.

**Table 2 animals-11-02967-t002:** Protocol used for free-range observations (ranging behaviour and vegetation) on Swedish organic broiler chicken farms. Free-range area not including winter garden.

Observation	Description
Free-ranging	
Birds outdoors	Estimate of total number of birds in free-ranging area
Bird dispersion	Estimate of proportion (%) of total number of birds outside at <5, 5 < 10, 10 < 15, 15 < 25 and ≥25 m, respectively
Maximum distance	Longest distance from the winter garden to where a bird was observed ranging
Free-range features	
Pasture: proportion of total free-range area	0 (very low: <20%); 1 (low: 20 < 40%); 2 (moderate: 40 < 60%); 3 (high: 60 < 80%); 4 (very high: ≥80%)
Vegetation cover: proportion of total free-range area ^1^	0 (none: 0%); 1 (extremely low: <5%); 2 (very low: 5 < 10%); 3 (low: 10 < 20%); 4 (moderate: 20 < 40%); 5 (high: 40 < 60%); 6 (very high: ≥60%)
Type of vegetation cover	Proportion (%) of total vegetation cover made up of bushes <100 cm, bushes ≥100 cm and trees, respectively
Artificial shelter	Description and number of objects, including dimensions as applicable, and an estimate of number of birds beneath

^1^ Vegetation cover defined as bushes and trees >50 cm.

**Table 3 animals-11-02967-t003:** Total number and median of event behaviours observed during behavioural observations (5 min per farm) in the rearing compartment on organic broiler farms (*n* = 8), and total number of farms on which each event behaviour was observed.

Behaviour	Observations of Behaviour (Total Counts)	Observations of Behaviour (Median and Range) per Farm	Total Number of Farms on Which Behaviour Was Observed
Gentle feather pecking	10	1 (0–5)	5
Squawks	8	0 (0–7)	2
Leg stretching	6	1 (0–2)	5
Perching	4 ^1^	1 (1–2)	3
Aggressive pecking	3	0 (0–2)	2
Wing stretching	2	0 (0–1)	2
Play-like activity	2	0 (0–2)	1
Fighting	1	0 (0–1)	1
Dust bathing	0	0 (0–0)	0
Flying	0	0 (0–0)	0
Other vocalisations	0	0 (0–0)	0
Severe feather pecking	0	0 (0–0)	0

^1^ Perching on water and feed lines.

**Table 4 animals-11-02967-t004:** Bird use of environmental enrichment items provided in the rearing compartment on organic broiler farms (*n* = 8): median number of birds on top and adjacent to item (unless only one observation per type of item) and total number of observations per type of item and farm.

Item	Number of Birds on Top of Item per Observation (Median and Range)	Number of Birds Adjacent to Item per Observation (Median and Range)	Total Number of Observations per Item	Number of Farms on Which Observations Were Performed (out of Total Farms Possible)
Perches ^1^	16 (13–21)	n/a	10	2 (2)
Cart ^2^	15	18	1	1 (2)
Pallet collar ^3^	7 (1–13)	7.5 (5–10)	2	2 (2)
Straw bale (large, round)	6 (4–8)	25 ^6^	4	1 (1)
Bales (small, square) ^4^	4 (0–6)	6 (2–11)	9	3 (3)
Plastic barrel upside down ^5^	2 (1–4)	9 (0–12)	5	1 (2)
Plastic bucket upside down	1	3	1	1 (1)
Eight steps-stepladder	0	4	1	1 (1)

^1^ Total length 21.6 m and 19.5 m, respectively. ^2^ Surface area ~1.5 m^2^. ^3^ On top = on pallet edge. ^4^ Straw, peat and lucerne. ^5^ Surface area ~0.4–1.4 m^2^. ^6^ Not possible to count exact number (i.e., estimate).

**Table 5 animals-11-02967-t005:** Total and median number of behaviours observed (per pop-hole and per farm) during group behavioural observations (5 min per pop-hole) at pop-holes (*n* = 23) on organic broiler farms (*n* = 8) and median number of observations per meter (width) pop-hole.

Behaviour	Observations of Behaviour (Total Counts)	Total Number of Pop-Holes at Which Behaviour Was Observed	Observations of Behaviour (Median and Range) per Pop-Hole	Total Number of Farms on Which Behaviour Was Observed	Observations of Behaviour (Median and Range) per Farm	Observations of Behaviour (Median and Range) per Meter (Width) Pop-Hole
Standing	243	22	10 (0–23)	8	23.5 (5–58)	6.5 (0–13.9
Sitting	108	19	3 (0–16)	8	7.5 (1–39)	1.5 (0–8)
Foraging	90	17	2 (0–14)	7	7 (0–31)	1.2 (0–8.5)
Preening	47	13	1 (0–8)	7	3.5 (0–13)	0.6 (0–4.9)
Turning back in	21	10	0 (0–7)	7	1.5 (0–9)	0 (0–3.5)
Running	20	8	0 (0–6)	5	1 (0–8)	0 (0–3)
Leg stretching	15	9	0 (0–3)	6	1.5 (0–5)	0 (0–1.5)
Walking (along pop-hole)	11	5	0 (0–3)	3	0 (0–5)	0 (0–1.8)
Gentle feather pecking	10	4	0 (0–5)	4	0.5 (0–5)	0 (0–2.5)
Wing stretching	6	6	0 (0–1)	5	1 (0–2)	0 (0–1)
Turning back out	6	6	0 (0–1)	3	0 (0–2)	0 (0–1)
Aggressive pecking	2	2	0 (0–1)	2	0 (0–1)	0 (0–0.6)
Dust bathing	2	2	0 (0–1)	2	0 (0–1)	0 (0–0.6)
Resting	1	1	0 (0–1)	1	0 (0–1)	0 (0–0.5)
Flying	0	0	0 (0–0)	0	0 (0–0)	0 (0–0)
Severe feather pecking	0	0	0 (0–0)	0	0 (0–0)	0 (0–0)
Fighting	0	0	0 (0–0)	0	0 (0–0)	0 (0–0)
Play-like activity	0	0	0 (0–0)	0	0 (0–0)	0 (0–0)

**Table 6 animals-11-02967-t006:** Proportion of broiler flocks observed free-ranging (FR) on Swedish organic chicken farms (*n* = 8), distance from winter garden and prevailing weather conditions (HR: heavy rain; D: drizzle; S: sunny; C: cloudy).

Farm	Proportion % of Flock FR ^1^	Distance (m) from Winter Garden (% of FR Chickens)					Maximum Distance (m) from Winter Garden	Temperature (°C)	Cloud cover	Precipitation	Average Wind Speed ^2^ (m/s)
		<5	5 < 10	10 < 15	15 < 25	≥25					
a	6 (250)	10	10	20	55	5	55	10.2	S	-	5 (9)
b	0 (0)	n/a	n/a	n/a	n/a	n/a	n/a	7.5	C	HR	5 (9)
c	0.2 (8)	37.5	0	0	62.5	0	19	12.2	C	D	6 (14)
d	1.1 (50)	20	60	10	10	0	18	16.8	S(C)	-	4 (8)
e	3.3 (160)	12.5	31.3	37.5	12.5	6.3	33	12.2	S	-	5 (10)
f	0 (0)	n/a	n/a	n/a	n/a	n/a	n/a	8.3	S	-	5 (13)
g	0.8 (40)	12.5	50	12.5	25	0	16	9.5	S	-	6 (14)
h ^3^	21.5 (215)	70	9.3	4.7	7	9.3	40	8.6	C	D	9 (19)

^1^ Absolute numbers in brackets. ^2^ Gusts (m/s) within brackets. ^3^ Observations performed at 08.00 h.

**Table 7 animals-11-02967-t007:** Description of fences around free-range areas on organic broiler farms (*n* = 8) in Sweden.

Farm	Description of Fence
a	Chicken wire (height 100 cm); buried 30 cm horizontally underground; slacking considerably in places
b	Sheep fence (height 100 cm); square openings 10–12 cm; slacking considerably in places
c	Wildlife fence (180–200 cm); ground-level electric fence
d	Sheep fence (100 cm); square openings 10–12 cm; slacking considerably in places; not enclosing entire range
e	Wildlife fence (180–200 cm)
f	Robust wildlife fence (155 cm); mink-proof fence bottom 100 cm, buried 20 cm underground; electric fence at top and ground level
g	Wildlife fence (180–200 cm)
h	Robust wire fence (height ~150 cm)

## Data Availability

Restrictions apply to the availability of these data. The data presented in this study is available on request from the corresponding author, with the permission of the participating farmers.

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
