# Peer review of "Behaviour in Slower-Growing Broilers and Free-Range Access on Organic Farms in Sweden"

_animals, 2021, doi:10.3390/ani11102967_

Round 1

Reviewer 1 Report

-in line 37, delet "poultry" 

-in line 177-185, I dont think in your research ADT results is appropriate, becaues its results is may be more relied on farmers's altitute to birds than rearing types. So researthers need to be very careful to present these results from different farms (it would be all right this results came from same farm in an appropriate research design). These results would also cause trouble when you discuss them. 

-The most results observed were from different farms which were varying in size of flocks, features in enrichmental environment, conditions of farms, climate and even experience of different farmers, I think you compare these results among farms like Table 6 may mean nothing in science.

-line 28--290, I think ADT results from your survey are difficult to compared with the previous studies, for example, you have no result of stocking density or leg heathy inspection, etc. 

Reviewer 2 Report

The manuscript is well written. The topic falls within the general scope of the Animals Journal,  and the study adds interesting information of organic broiler behaviour.

The main question addressed by the research is knowledge of bird welfare in slower-growing hybrids under commercial settings.

The topic is original and relevant in the field because welfare in organic chicken is influenced by various factors and the studies conducted so far do not always appear complete.

Despite the limited number of farms and the lack of repeated observations, the results from the bird observations and from farmer interviews provide novel insights into organic broiler production on Swedish farms.

Much of the research reported on welfare in organic chicken is it was carried out on small groups in experimental tests. The aforementioned research, on the other hand, describes the current situation and practical solutions applied to organic broiler farms in Sweden.

Each farm was visited during one day. The authors could make more visits.

The conclusions are consistent with the evidence and arguments presented and do they address the main question posed.

The references are appropriate.

The tables reported several event behaviours observed during behavioural observations. on the other hand, the protocol was used for free-range  observations on Swedish organic broiler chicken farms.

It is acceptable for publication in its present form.

Reviewer 3 Report

It would be very important to describe productive cahracteristics of genotypes studied (Rowan Ranger and Hubbard JA57/Hubbard JA87), but more important would be to associate specific behaviours with genotypes as far as authors can do.   Productive performance may be related to behavior
